# Concentrations, Distribution, and Pollution Assessment of Metals in River Sediments in China

**DOI:** 10.3390/ijerph18136908

**Published:** 2021-06-27

**Authors:** Guoqi Lian, Xinqing Lee

**Affiliations:** 1The State Key Laboratory of Environmental Geochemistry, Institute of Geochemistry, Chinese Academy of Sciences, Guiyang 550081, China; lee@mail.gyig.ac.cn or; 2College of Resources and Environment, University of Chinese Academy of Sciences, Beijing 100049, China; 3School of Chemistry and Materials Engineering, Liupanshui Normal University, Liupanshui 553004, China; 4Guizhou Provincial Key Laboratory of Coal Clean Utilization, Liupanshui 553004, China

**Keywords:** metals, distribution, pollution assessment, river sediments, China

## Abstract

This study conducted a review on the concentrations, spatial distribution and pollution assessment of metals including As, Hg, Cd, Co, Cr, Cu, Mn, Ni, Pb and Zn in 102 river sediments in China between January 2008 and July 2020 based on the online literature. The geo-accumulation index (*Igeo*) and potential ecological risk index (*RI*) were used for the pollution assessment of the metals. The results showed that the ranges of metals were: 0.44 to 250.73 mg/kg for As, 0.02 to 8.67 mg/kg for Hg, 0.06 to 40 mg/kg for Cd, 0.81 to 251.58 mg/kg for Co, 4.69 to 460 mg/kg for Cr, 2.13 to 520.42 mg/kg for Cu, 39.76 to 1884 mg/kg for Mn, 1.91 to 203.11 mg/kg for Ni, 1.44 to 1434.25 mg/kg for Pb and 12.76 to 1737.35 mg/kg for Zn, respectively. The median values of these metals were descending in the order: Mn > Zn > Cr > Cu > Pb > Ni > Co > As > Cd > Hg. Compared with the SQGs, As and Cr manifested higher exceeding sites among the metals. Metals of river sediments manifested a significant spatial variation among different regions, which might be attributed to the natural weathering and anthropogenic activity. The mean *I_geo_* values of the metals presented the decreasing trends in the order: Cd > Hg > Zn > Cu > As > Pb > Ni > Co > Cr > Mn. Cd and Hg manifested higher proportions of contaminated sites and contributed most to the RI, which should be listed as priority control of pollutants. Southwest River Basin, Liaohe River Basin, and Huaihe River Basin manifested higher ecological risks than other basins. The study could provide a comprehensive understanding of metals pollution in river sediments in China, and a reference of the control of pollutant discharge in the river basins for the management.

## 1. Introduction

With the rapid growth of population, and industrial and agricultural development, heavy metal pollution in aquatic systems has gradually become a global issue and attracted widespread attention of studies [1,2]. In the aquatic environment, river sediments serve as the repository of metals [3], and are considered as one of the important monitoring indicators for long-term metal deposition pollution in ecosystems [4,5]. Sediments may directly affect overlying waters and become a potential secondary non-point source of metals pollution [3,6]. Due to the persistent, toxic, less degradable nature and bioaccumulation of metals in the environment, they could be released under favorable conditions such as redox potential, pH, dissolved oxygen, and temperature, and pose a great potential threat to aquatic ecosystems and the local inhabitants through the food chain [2,5,7,8,9]. Metals in river sediments originate from both natural sources and anthropogenic activities, such as chemical leaching of bedrock, water drainage, mining, the discharge of urban industrial and rural agricultural wastewaters [10,11,12]. Studying the content, distribution, and harm of metals in river sediments can be helpful to better understand the impact of human activities on river ecosystem.

In China, heavy metal pollution has become one of the main concerns of the government; several metals including Hg, Cr, Cd, Pb and As have been listed as the target of total load control in the 12th Five-Year Plan of Environmental Protection [13]. It was reported that 40% of the 10 primary river systems monitored in China had been disturbed by human factors and resulted in adverse impacts to human beings [14]. Historically, there have been serious water pollution problems in China’s rivers, such as the water pollution accident in the Tuojiang River of Sichuan province in 2004, the major water pollution incident in the Songhua River in 2005, arsenic contamination in Yueyang of Hunan province in 2006, and arsenic pollution in the Huai River basin in 2008, etc. These major water pollution accidents have caused a great impact on people’s life; furthermore, they generated long-term influence that was difficult to eliminate from the aquatic ecosystem.

A large number of studies have been conducted in metals contamination assessment in river sediments in different regions of China. For example, Zhao, Ye [3] carried out heavy metal contamination assessment in river sediments of the Pearl River Delta; Chai, Li [15] assessed metals distribution, contamination, and ecological risks in the surface sediments of the Xiangjiang River; Cheng, Wang [5] assessed heavy metal contamination in the sediments of the Yellow River Wetland National Nature Reserve; Yang, Chen [16] reported the heavy metal contents and ecological risk in sediments in the Wei River Basin, etc. Previous studies have shown that anthropogenic activities have caused metals contamination in the river sediments. However, these studies were mainly focused on river sediments in certain river basin or individual rivers; to our knowledge, there is currently no systematic, comprehensive assessment and comparison of the pollution status of metals in river sediments in China. Therefore, it is particularly urgent to investigate the spatial distribution, pollution degree and ecological risks of metals in river sediments in different regions of China on a national scale. The potential ecological risk index (RI) is usually used as an indicator to assess the risk of heavy metal contamination in river and lake sediments [17].

This study was conducted to evaluate the pollution status of metals (As, Hg, Cd, Co, Cr, Cu, Mn, Ni, Pb and Zn) in river sediments in different basins of China based on the online literature. The major objectives of this study are: (i) to analyse the concentration and spatial distribution of metals in river sediments in different basins of China; (ii) to evaluate the pollution degree of metals by using the geoaccumulation index (*I_geo_*); (iii) to assess the potential ecological risks caused by the metals based on the potential ecological risk index (RI); (iv) to investigate the pollution status and risks of metals in different river basins. The results can provide scientific support for environment management to develop corresponding control measures, and it will generate beneficial impacts on aquatic ecosystems and human health.

## 2. Methods

### 2.1. Searching Method

This study conducted a comprehensive search of literature on metals in the sediments of 102 rivers in China published from January 2008 to July 2020, gained from the China National Knowledge Infrastructure (CNKI) and Web of Science using the terms “heavy metal”, “metals”, “China”, “river”, and “sediment” as the searching subjects. A total of 2182 and 2810 articles were obtained from CNKI and Web of Science, respectively. Finally, we selected 102 pieces of literature (18 in Chinese and 84 in English) by eliminating the articles that were irrelevant and unable to provide total contents of the metals in the sediment, through three screening procedures including title review, abstract review and full text review (Figure 1); a total of 3063 samples sites were contained in this study. The metals were digested with mixed acids HNO_3_ + HF + HCl or HF + HClO_4_ + HNO_3_; the total concentrations of the metals were determined by ICP-MS, ICP-OES, ICP-AES or AAS. The sample analysis process was carried out in strict accordance with the standards. The mean contents of metals, research area, river name, published year and sampling number of the selected articles were extracted and recorded for further statistical analysis (Table 1), and the distribution of the river basins were presented in Figure 2. The sample sites in different river basins were distributed as follows: Heilongjiang River Basin (*n* = 266), Liaohe River Basin (*n* = 395), Haihe River Basin (*n* = 321), Yellow River Basin (*n* = 210), Huaihe River Basin (*n* = 174), Yangtze River Basin (*n* = 839), Southeast Coastal Basin (*n* = 82), Pearl River Basin (*n* = 615), Southwest River Basin (*n* = 102), and Northwest River Basin (*n* = 59).

### 2.2. Analytical Methods

#### 2.2.1. Sediment Quality Guidelines

Sediment quality guidelines (SQGs) proposed by MacDonald, Ingersoll [104] were used to evaluate the quality of the sediment in the freshwater ecosystems and determine the degree to which the metals of the sediment might pose a threat to the aquatic organisms. It contains threshold effect concentration (TEC) and a probable effect concentration (PEC). When values are below the TEC, harmful effects are unlikely to be observed; values above PEC indicate that harmful effects are likely to be observed.

#### 2.2.2. Geoaccumulation Index

The geo-accumulation index (*I_geo_*) was used to quantify metals contamination caused by both natural geological and geographical processes and human activities [1], which was introduced by Müller [105]. The *I_geo_* values were calculated by the following equation:(1)$$Igeo=\log_{2}\left[\frac{Cn}{1.5Bn}\right]$$
where *C_n_* represents the concentration of the measured metal (*n*) in the sediment and *B_n_* represents the regional geochemical background (BG) value of the metal; we chose the arithmetic means of background values of the metals in soils in different provinces of China as the BG regulated by Chinese environmental monitoring stations [106]; the factor 1.5 is the background matrix correction factor. The *I_geo_* can be classified into seven classes: class 0 (*I_geo_* ≤ 0), uncontaminated; class 1 (0 < *I_geo_* ≤ 1), uncontaminated to moderately contaminated; class 2 (1 < *I_geo_* ≤ 2), moderately contaminated; class 3 (2 < *I_geo_* ≤ 3), moderately to heavily contaminated; class 4 (3 < *I_geo_* ≤ 4), heavily contaminated; class 5 (4 < *I_geo_* ≤ 5), heavily contaminated to extremely contaminated; class 6 (5 ≤ *I_geo_*), extremely contaminated.

#### 2.2.3. Potential Ecological Risk Index

The potential ecological risk index (*RI*) proposed by Hakanson [107] was employed to quantify the level of ecological risk degree of metals in aquatic sediments [85]. It is widely used by researchers as an effective method to assess the contamination levels and potential risks of heavy metal in the sediments by combining ecological and environmental effects with toxicology. The calculation of *RI* is based on the following equation:(2)$$E_{r}^{i}=T_{r}^{i}\times C_{f}^{i}=T_{r}^{i}\times \frac{C_{s}^{i}}{C_{n}^{i}}$$
(3)$$RI=\sum_{i=1}^{n}E_{r}^{i}$$
where $T_{r}^{i}$ is the biological toxic response factor for heavy metal *i*, the toxic response factors of metals are: As = 10, Hg = 40, Cd = 30, Co = Cu = Ni = Pb = 5, Cr = 2, and Mn = Zn = 1 [16,85], $C_{f}^{i}$ refers to the toxic response factor of the heavy metal *i*, $C_{s}^{i}$ is the measured concentration of the heavy metal *i*, $C_{n}^{i}$ is the background concentration of the heavy metal *i*. The classifications of *RI* values are as follows [107]: *RI* < 150 (low ecological risk), 150 ≤ *RI* < 300 (moderate ecological risk), 300 ≤ *RI* < 600 (considerable ecological risk), and *RI* > 600 (very high ecological risk), respectively.

### 2.3. Statistical Analysis

Microsoft Office 2016 for Windows (Microsoft office, Washington, DC, USA) was applied to perform all statistical analysis. A scatter diagram was performed to map the spatial distribution of metals in rivers sediments selected in China based on Origin (Version 2018) (OriginLab, Massachusetts, MA, USA).

## 3. Results and Discussion

### 3.1. Concentrations and Spatial Distribution of Metals

#### 3.1.1. Heavy Metal Concentrations

The concentrations of 10 metals in selected river sediments were summarized in Table 2. The ranges of metals were: 0.44 to 250.73 mg/kg for As, 0.02 to 8.67 mg/kg for Hg, 0.06 to 40 mg/kg for Cd, 0.81 to 251.58 mg/kg for Co, 4.69 to 460 mg/kg for Cr, 2.13 to 520.42 mg/kg for Cu, 39.76 to 1884 mg/kg for Mn, 1.91 to 203.11 mg/kg for Ni, 1.44 to 1434.25 mg/kg for Pb and 12.76 to 1737.35 mg/kg for Zn, respectively. The median values of these metals were descending in the order: Mn > Zn > Cr > Cu > Pb > Ni > Co > As > Cd > Hg. The variable coefficient (CV) of the metals presented in the trends: Pb > Hg > Cd > Co > As > Cu > Zn > Cr > Ni > Mn, and Mn, Zn, and Pb showed higher standard deviation (SD) among the metals. When comparing with the mean values of these metals with the background values of China, all the metals were larger than the BG values, suggesting that anthropogenic activities exert great influences on the river sediments, especially Cd and Hg, which were 29.59 and 11.54 times higher than BG values. Compared with the SQGs, the mean contents of the metals were between the range of TEC and PEC, while some of the sites exceeded the PEC; the exceeding rates were 21.82% in As, 15.38% in Hg, 16.67% in Cd, 20.45% in Cr, 7.29% in Cu, 19.70% in Ni, 9.09% in Pb and 16.32% in Zn, and the values of these metals between TEC and PEC accounted for 52.73%, 38.46%, 28.57%, 65.91%, 61.46%, 54.55%, 43.43% and 38.78%, respectively, indicating that these sites probably cause adverse biological effects. As and Cr manifested the higher exceeding sites among the metals, which most likely generate adverse effects in the ecosystem, followed by Ni, Cd, Zn, Hg and Pb, the least was Cu.

In addition, when comparing the mean values of the metals with other countries (Table 3), As, Cd and Co were higher than the those of other selected countries, Hg was higher than Awash River in Ethiopia, Cr, Cu, Mn, Ni, Pb and Zn were larger than in most other countries, and lower than a few countries; e.g., Cr was lower than Khorramabad River in West Iran and Awash River in Ethiopia, but was higher than those of other selected countries, Pb was lower than Tinto River in Spain and was over the other selected countries, Zn was below Danube River in Serbia, Tinto River in Spain and Awash River in Ethiopia, while it was higher than Red River in Vietnam, Merang river in Malaysia, Georges River in Australia and River Chenab in Pakistan.

#### 3.1.2. Spatial Distribution

The spatial distributions of metals were presented in Figure 3. As was illustrated in Figure 3, the spatial variations of the concentrations of metals in the river sediments varied significantly. High As levels were mainly distributed in Lanmuchang, Tributary of Zhedong River and Hengyang Segment of Xiangjiang River Basin; the concentrations of these sites were 250.73 mg/kg, 168.72 mg/kg and 135.2 mg/kg, respectively. High Hg levels were mainly located in Wuli River, Shuangqiao River and Yarlung Tsangpo River, with the highest Hg content of 8.67 mg/kg in Wuli River sites. High Cd levels were mainly observed in Tuohe River (40 mg/kg), Hengyang Segment of Xiangjiang River Basin (21.66 mg/kg), Wenruitang River (13.84 mg/kg) and Xiangjiang River (13.68 mg/kg). Co showed higher concentrations in Tuo River of Suzhou (251.58 mg/kg) and Urban River in Northern Anhui Province (245.64 mg/kg). Cr exhibited higher levels in Ziya River (460 mg/kg), Wenruitang River (369.11 mg/kg), Kuye River (289.59 mg/kg), Xiaoqinghe Watershed (257.79 mg/kg) and Lanmuchang (201.41 mg/kg). Higher Cu sites were situated in Wenruitang River (520.42 mg/kg), Urban Rivers in Baoan District in Shenzhen (465.91 mg/kg), Shuangqiao River (435.20 mg/kg) and Guangzhou Section of the Pearl River (351.80 mg/kg). A higher Mn was distributed in Hengyang Segment of Xiangjiang River Basin (1884 mg/kg), Xiangjiang River (1805.17 mg/kg), Sanmenxia Section of Yellow River Wetland National Nature Reserve (1633.50 mg/kg), Kuye River (1471.91 mg/kg), Zijiang River (1322.89 mg/kg), Jinjiang River Estuary (1264 mg/kg) and Jiulong River (1132.90 mg/kg). Ni levels were high in Wenruitang River, Shenzhen River and Lianshui River, with the concentrations of 203.11 mg/kg, 120 mg/kg and 102 mg/kg, respectively. Pb had the highest concentration in Taizihe River, with the content of 1434.25 mg/kg, followed by 906.50 mg/kg in the river in the Baiyinnuoer lead–zinc Mining Area. Zn was higher in Taizihe River, the river in the Baiyinnuoer lead–zinc Mining Area, Lianshui River, Wenruitang River, and Nanfei River in Chaohu Basin, with concentrations of 1737.35 mg/kg, 1432.88 mg/kg, 1299 mg/kg, 1065.82 mg/kg and 869.30 mg/kg, respectively. On the whole, each single metal of river sediments manifested a significant spatial variation among different regions, which might be attributed to the natural weathering and anthropogenic activity that caused pollutants to enter into aquatic systems, eventually accumulating in sediments.

### 3.2. Contamination Assessment of Metals

The *I_geo_* values and proportions of *I_geo_* classifications of the metals of the river sediments in this study were presented in Figure 4. As illustrated in Figure 4, the *I_geo_* values of the metals ranged as follows: As ranged from −1.99 to 5.93, with the mean value of 0.23; Hg ranged from −1.73 to 7.29, with the mean value of 1.45; Cd ranged from −4.48 to 8.10, with the mean value of 2.65; Co varied from −4.83 to 3.73, with the average being −0.27; Cr varied from −4.78 to 2.25, with the average being −0.30; Cu varied from −4.43 to 4.86, with the average being 0.51; Mn ranged between −4.75 and 1.45, with a mean value of −0.57; Ni ranged between −4.87 and 2.92, with a mean value of −0.24; Pb ranged between −4.46 and 5.48, with a mean value of 0.14; Zn ranged between −3.30 and 4.30, with a mean value of 0.67. The mean *I_geo_* values of the metals presented decreasing trends in the order: Cd > Hg > Zn > Cu > As > Pb > Ni >Co > Cr > Mn. The mean *I_geo_* values of Ni, Co, Cr and Mn were below 0, belonging to class 0 and indicating uncontaminated grade; most sites of these metals exhibited uncontaminated levels, while some sites of these metals showed certain proportions of pollution, such as Cr and Ni. The proportions of uncontaminated to moderately contaminated, moderately contaminated and moderately to heavily contaminated sites were 23.86%, 5.68% and 4.55% for Cr, and 21.21%, 9.09% and 4.55% for Ni, respectively. The mean *I_geo_* value of Cd reached 2.65, which was the highest among the metals and belonged to class 3, suggesting moderately to heavily contaminated grade, which presented 11.90% uncontaminated sites, 13.10% uncontaminated to moderately contaminated sites, 19.05% moderately contaminated sites, 11.90% moderately to heavily contaminated sites, 17.86% heavily contaminated sites, 8.33% heavily contaminated to extremely contaminated sites and 17.86% extremely contaminated sites, respectively. Hg contamination reached moderately contaminated level ranking class 2, with the sites of uncontaminated, uncontaminated to moderately contaminated, moderately contaminated, moderately to heavily contaminated, heavily contaminated, heavily contaminated to extremely contaminated and extremely contaminated attributing for 30.77%, 25.64%, 10.26%, 5.12%, 12.82%, 7.69% and 7.69%, respectively. The mean contamination of As, Cu, Pb and Zn were ranked as class 1, indicating uncontaminated to moderately contaminated grade, while parts of sites were extremely contaminated by As and Pb; the proportions were 1.82% in As and 2.04% in Pb, respectively. Cu and Zn showed heavily contaminated to extremely contaminated sites, constituting 2.08% and 2.04%, respectively. In general, Cd and Hg manifested higher proportions of heavily contaminated to extremely contaminated and extremely contaminated sites, which could be regarded as dominant pollutants among the metals in river sediments in this study.

As illustrated in Figure 5a, the average $E_{r}^{i}$ values showed the following trend: Cd(910.76) > Hg(750.41) > As(40.14) > Cu(18.57) > Pb(18.38) > Co(12.59) > Ni(8.94) > Zn(4.05) > Cr(3.25) > Mn(1.39). Cd and Hg contributed most to the RI (Figure 5b), the mean contribution proportions of which were 69.78% and 38.88%, respectively. It could be speculated that although the contents of Cd and Hg were not very high, they posed a relatively higher potential ecological risk among the metals in river sediments in this study, which might be related with the low background values and strong toxicity coefficient of these metals, which resulted from the existing form of the metals in the sediments such as the easy dissolution and transport of a major chemical form of Cd in the sediments [85], which should be paid much more attention. On the contrary, the contents of Mn and Zn were very high, while they posed very low ecological risks, and Mn showed the lowest potential ecological risk. In comparison, the $E_{r}^{i}$ values of As, Cu, Pb, Co, Ni, Zn, Cr and Mn were below 150, which posed lower ecological risks, and the mean contribution rates of these metals to RI were below 10%. However, the $E_{r}^{i}$ values of As in Jiaozhou Bay Catchment reached 916.67. Pb in the river in the Baiyinnuoer lead–zinc Mining Area was 302.17; in addition, due to the accumulation and toxicity of the metals, these metals should also be given certain attention.

As illustrated in Figure 5a and Figure 6, the RI values ranged from 7.53 to 12,388.05, with the mean value being 1110.63, suggesting a very high ecological risk level. According to the RI classification standard, 32 sites were in the low ecological risk, 18 sites belonged to moderate ecological risk, 16 sites were classified into considerable ecological risk, and 36 sites were classified into very high ecological risk, the proportions of which being 31.37%, 17.65%, 15.69% and 35.29%, respectively.

The spatial distribution of $E_{r}^{i}$ of Cd and Hg and RI in different river sediments in this study was shown in Figure 6. The high Cd ecological risks sites (Figure 7a) were mainly distributed in Hengyang segment of Xiangjiang River Basin, Tuo River of Suzhou, Lianshui River, Tuohe River, Lianshan River, Quannan Section of Taojiang River, Shiqiao River, Shawan River, Nanfei River in Chaohu Basin, Shuangqiao River, Longjiang River, Xiashan Stream, Xiangjiang River, Wenruitang River, Wuli River, Xiaoqinghe Watershed and the river in the Baiyinnuoer lead–zinc Mining Area. For Hg (Figure 7b), the high ecological risks sites were mainly located in Hengyang Segment of Xiangjiang River Basin, Nanfei River in Chaohu Basin, Xiashan Stream, Wuli River, Lianshan River, Rivers of Chaohu City, Quannan Section of Taojiang River, Shuangqiao River, Songhua River Harbin Region, Yarlung Tsangpo River, Rivers in Beijing Central District and Songhua River. In general, the river sites such as Hengyang Segment of Xiangjiang River Basin, Nanfei River in Chaohu Basin, Xiashan Stream, Wuli River, Lianshan River, Quannan Section of Taojiang River and Shuangqiao River should be paid special concern, as these sites manifested both Cd and Hg, indicating high ecological risks.

From Figure 7c, the RI values in some river sediments exceeded 600, which presented very high ecological risks, such as Hengyang Segment of Xiangjiang River Basin, Shaying River, Lianshui River, Tuohe River, Nanfei River in Chaohu Basin, Xiashan Stream, Wuli River, Lianshan River, Quannan Section of Taojiang River, Shiqiao River, Shawan River, Shuangqiao River, Longjiang River, Xiangjiang River, Yarlung Tsangpo River, Wenruitang River, Xiaoqinghe Watershed, the river in the Baiyinnuoer lead–zinc Mining Area and Songhua River, etc., which were mainly distributed in the eastern regions. The eastern coastal cities of China have high population densities, relatively developed economies and active industrial activities, such as chemical, electronic processing, metal equipment manufacturing, prevention, leather and other industrial waste emissions, which are associated with heavy metal emission and may be the main causes of heavy metal pollution in eastern Chinese rivers [117]. The high risks of the metals in river sediments might be primarily caused by the anthropogenic activities brought about by the economic development, such as the sewage discharged by industrial and domestic activities, and the agrochemical usage including fertilizers and herbicides [10]. For example, the high Cd and Hg in Quannan section of the Taojiang River were mainly affected by aquaculture, mining and smelting [94]. The severe potential ecological risk in the Xiangjiang River was associated with long-term mining and smelting nonferrous metals activities [15].

### 3.3. Pollution Assessment in Different River Basins

The box plots of *I_geo_* and RI values in different river basins were presented in Figure 8. From Figure 8, the mean *I_geo_* values of Hg and Cd were higher than other metals in Heilongjiang River Basin, Liaohe River Basin, Haihe River Basin and Yellow River Basin, which were 2.86 and 2.11, 4.09 and 2.73, 1.91 and 1.02, and 2.75 and 1.79 in these river basins, respectively, indicating different degrees of pollution. The mean *I_geo_* values of Cd in Huaihe River Basin, Pearl River Basin, Southeast Coastal River Basin and Yangtze River Basin were highest among the metals, which were 2.29, 2.86, 3.62 and 3.51, respectively, belonging to moderately to heavily contaminated, moderately to heavily contaminated, heavily contaminated and heavily contaminated levels, respectively. In the Southwest River Basin, the mean *I_geo_* values of Hg and As were higher than other metals, which were 3.85 and 0.99, respectively, and could be classified into heavily contaminated and uncontaminated to moderately contaminated levels, respectively. In addition, all the mean *I_geo_* values of the metals in the Northwest River Basin were below 0, manifesting uncontaminated levels. It could be speculated that Hg and Cd were the most important pollutant in the sediments of river basins in China in this study, and should be listed as priority pollutants for future contamination control. Additionally, the sample points selected in each basin were unbalanced; for example, the points in Southeast Coastal River Basin, Southwest River Basin and Northwest River Basin were too few; therefore, it may not well reflect the pollution status in these basins.

As shown in Figure 8, the mean RI values in the Heilongjiang River Basin, Haihe River Basin and Northwest River Basin were 489.24, 246.90 and 71.06, respectively, suggesting considerable ecological risk, moderate ecological risk and low ecological risk, respectively. The mean RI values in other basins were above 600, indicating very high ecological risk, and the higher RI values were mainly located in the Southwest River Basin (1947.56), Liaohe River Basin (1940.05) and Huaihe River Basin (1365.69).

## 4. Conclusions

In this study, the concentrations, spatial distribution and pollution assessment of metals including As, Hg, Cd, Co, Cr, Cu, Mn, Ni, Pb, and Zn in river sediments in China were analyzed and reviewed. The concentrations of the 10 metals manifested that anthropogenic activities exert great influences on the river sediments. It showed a significant spatial variation among metals in different regions. The *I_geo_* values suggested that most sites of As, Co, Cr, Mn, Ni and Pb were uncontaminated, while the highest proportions of contamination were Hg, Cd, Cu, and Zn. Among these metals, Cd and Hg contributed most to the potential ecological risk in the river sediments, which could be regarded as dominant pollutants among the metals in this study.

The mean RI values suggested a very high ecological risk level of the river sediments; the proportions of very high ecological risk reached 35.29% in the river sediments sites. Southwest River Basin, Liaohe River Basin and Huaihe River Basin manifested higher ecological risk than other basins. The study provided certain theoretical basis to strengthen the management of pollutant discharge in the river basins, in particular for the control of toxic and harmful metals such as Hg, Cd, and As. Furthermore, detailed investigation and research on pollution sources in different river basins should be carried out and pollution control measures formulated according to different basins in future studies.

## Figures and Tables

**Figure 1 ijerph-18-06908-f001:**
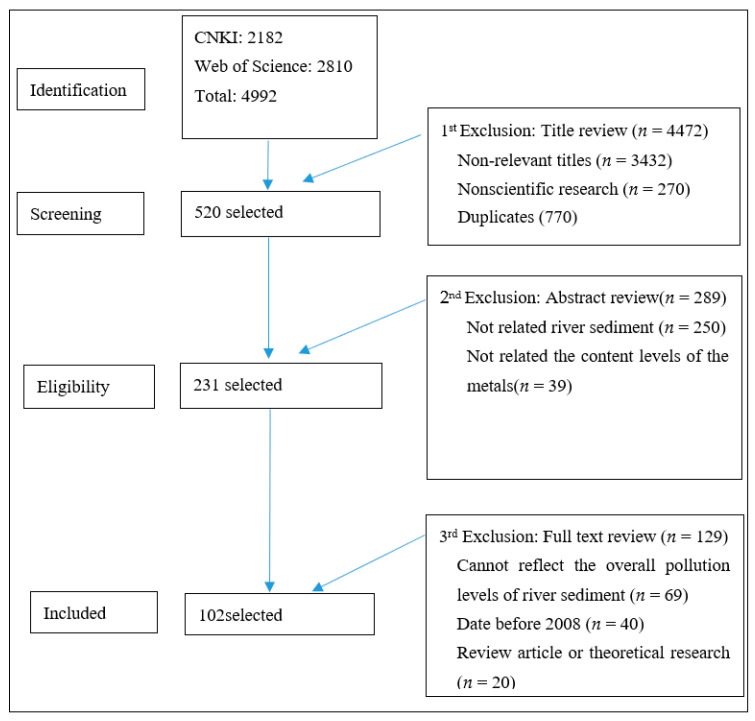
Flowchart for searching and screening procedure.

**Figure 2 ijerph-18-06908-f002:**
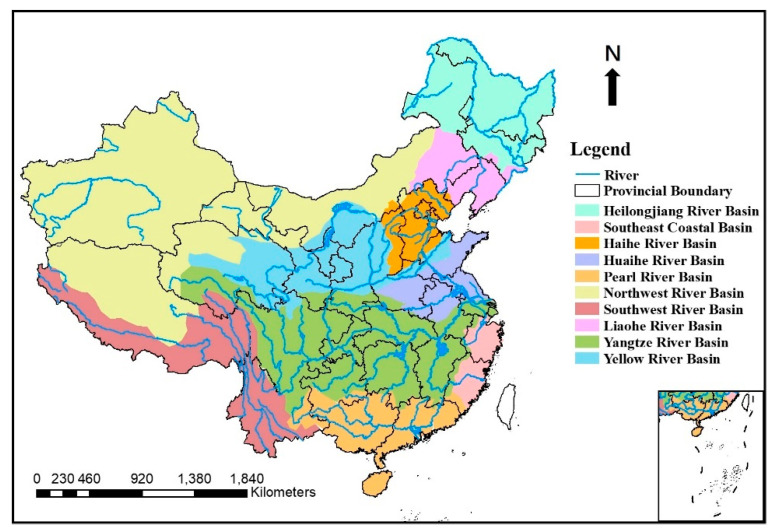
The distribution of the river basins.

**Figure 3 ijerph-18-06908-f003:**
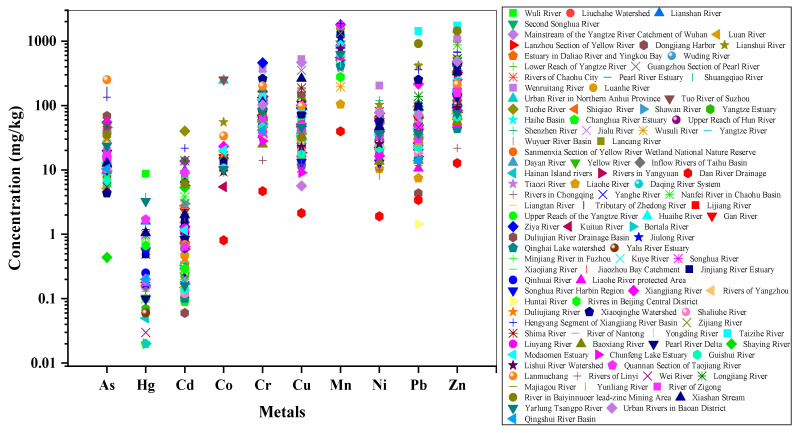
Spatial distribution of metals in river sediments in this study.

**Figure 4 ijerph-18-06908-f004:**
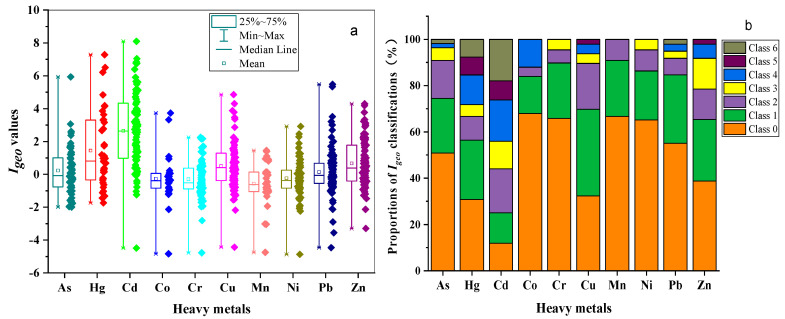
Box plots of *I_geo_* values (**a**) and proportions of *I_geo_* classifications of the metals (**b**).

**Figure 5 ijerph-18-06908-f005:**
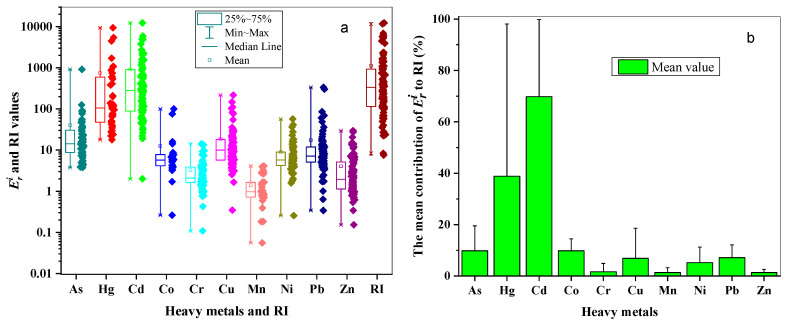
Box plots of $E_{r}^{i}$ and RI values (**a**) and the mean contribution of $E_{r}^{i}$ to RI of each metal (**b**).

**Figure 6 ijerph-18-06908-f006:**
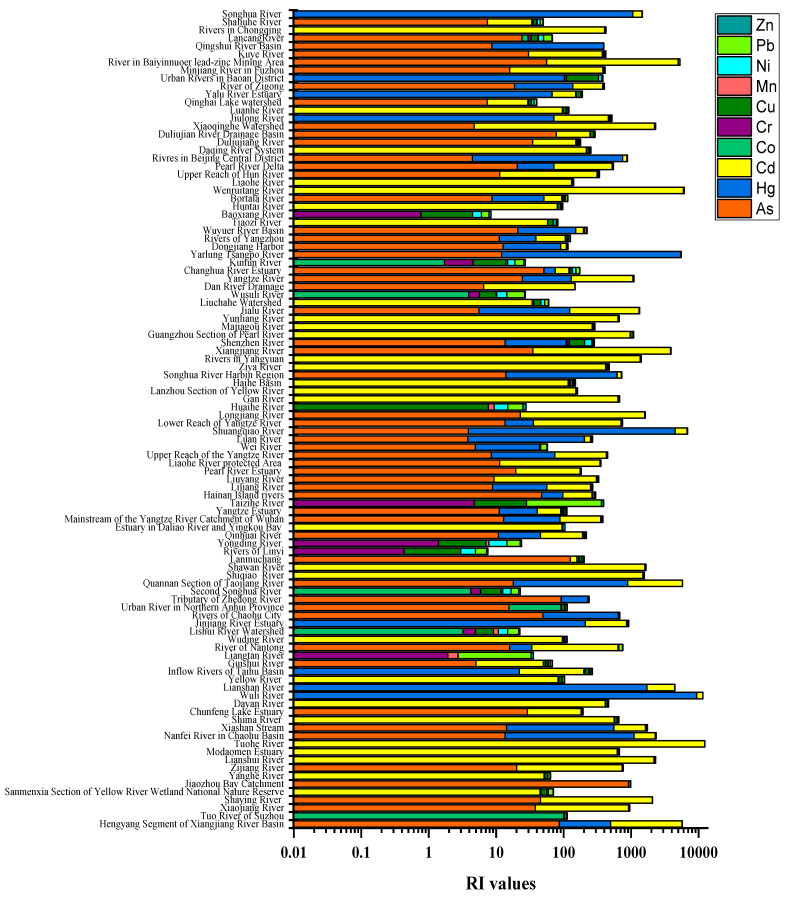
Values of the potential ecological risk index (RI) for metals in the river sediments.

**Figure 7 ijerph-18-06908-f007:**
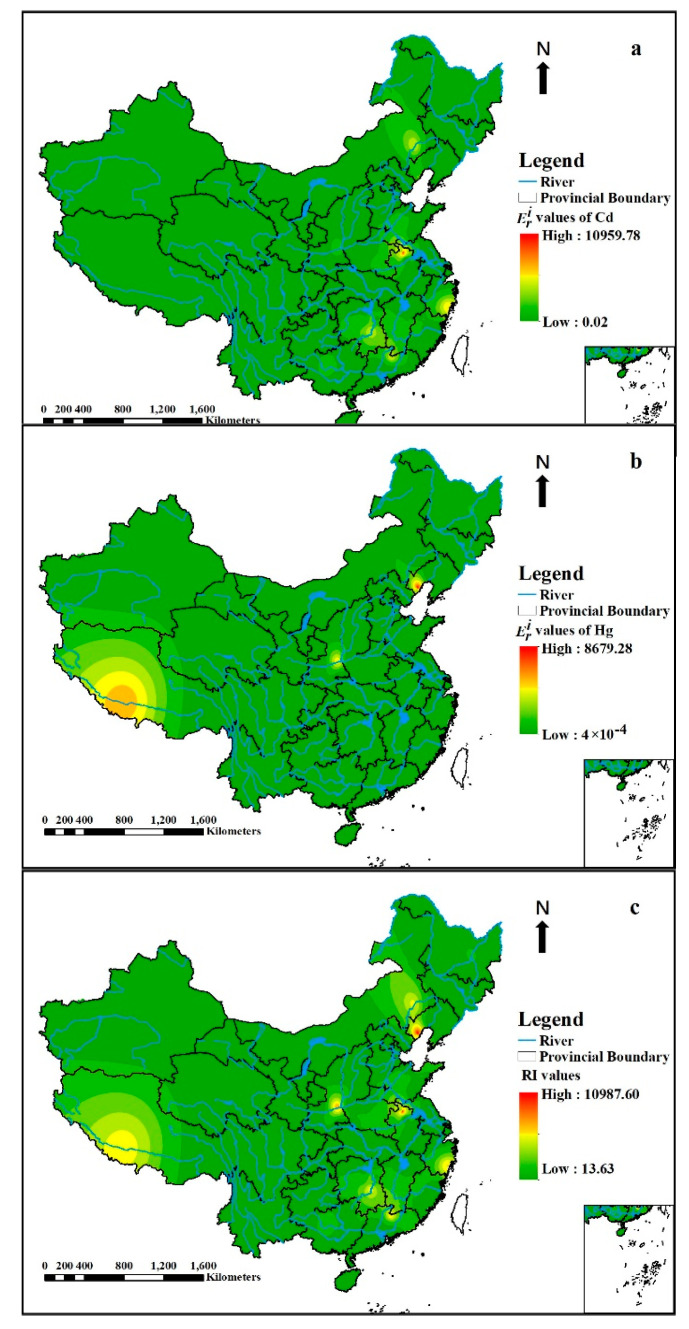
The spatial distribution of $E_{r}^{i}$ of Cd (**a**) and Hg (**b**) and RI (**c**) in river sediments in this study.

**Figure 8 ijerph-18-06908-f008:**
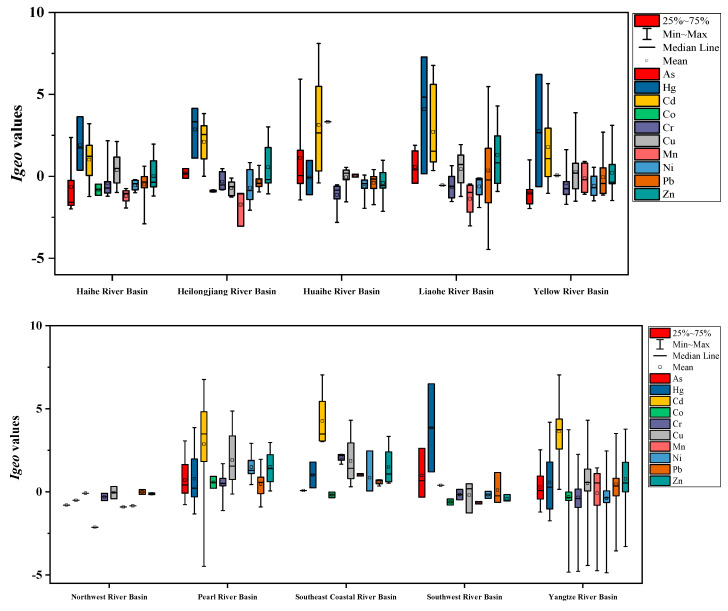
The box plots of *Igeo* and RI values in different river basins.

**Table 1 ijerph-18-06908-t001:** Statistics of chemical parameters of metals in river sediments in China (μg/L).

River Sites	Year	Number of the Sites	As	Hg	Cd	Co	Cr	Cu	Mn	Ni	Pb	Zn	Reference
Wuli River	2008	10	—	8.67	7.95	—	—	56.63	—	—	80.50	525.20	[18]
Liuchahe Watershed	2008	—	—	—	0.11	—	67.80	39.80	—	38.30	37.70	53	[19]
Lianshan River	2008	4	—	1.59	9.73	—	—	73.08	—	—	104.90	450.90	[18]
Second Songhua River	2008	—	—	—	—	9.97	49.42	23.58	775	18.75	23.76	90.29	[6]
Mainstream of the Yangtze River Catchment of Wuhan	2009	16	15.85	0.15	1.53	—	87.82	51.64	—	41.91	45.18	140.27	[20]
Luan River	2009	10	5.15	0.18	0.15	—	71.47	45.98	—	—	22.11	75.52	[21]
Lanzhou Section of Yellow River	2010	—	—	—	0.59	—	—	18.49	—	—	16.13	78.53	[22]
Dongjiang Harbor	2010	11	17.13	0.07	0.06	—	—	19.36	—	—	4.34	88.40	[23]
Lianshui River	2011	64	—	—	9.11	55	95	71	—	102	412	1299	[24]
Estuary in Daliao River and Yingkou Bay	2011	35	—	—	0.34	—	60.83	—	615.58	35.23	—	—	[25]
Wuding River	2011	5	—	—	0.30	—	60.71	19	426.86	29.64	15.60	76.55	[26]
Lower Reach of Yangtze River	2011	83	13.54	0.16	2.82	17.33	98.32	48.61	—	41.49	50.77	129.73	[27]
Guangzhou Section of Pearl River	2011	22	—	—	1.80	—	97.40	351.80	—	—	103.50	387.40	[28]
Rivers of Chaohu City	2012	9	44.35	0.49	—	—	102.03	79.44	—	—	49.46	206.07	[13]
Pearl River Estuary	2012	20	17.42	—	0.29	—	—	—	—	—	40.51	109.09	[29]
Shuangqiao River	2012	66	4.40	3.80	7.10	—	52.70	435.20	—	—	—	779.90	[30]
Wenruitang River	2012	29	—	—	13.84	—	369.11	520.42	—	203.11	58.68	1065.82	[31]
Luanhe River	2012	15	—	—	0.30	—	60.40	48.07	—	26.15	25.55	76.42	[32]
Urban River in Northern Anhui Province	2013	37	14	—	—	245.64	53.42	44.80	854	—	33.45	107.43	[33]
Tuo River of Suzhou	2013	5	—	—	—	251.58	55.51	44.91	—	—	—	108.47	[34]
Tuohe River	2013	—	—	—	40	—	46	35	853	—	24	61	[35]
Shiqiao River	2013	9	—	—	2.79	—	133	100	—	66	96	327	[12]
Shawan River	2013	7	—	—	2.99	—	109	75	—	53	86	253	[12]
Yangtze Estuary	2013	—	10.10	0.07	0.23	—	86	29	—	34	27	93	[36]
Haihe Basin	2013	117	—	—	0.36	13.40	81.90	53.30	435	27.80	20	256	[37]
Changhua River Estuary	2013	27	9.50	0.02	0.09	—	53.10	15	—	23	27	73.70	[38]
Upper Reach of Hun River	2013	—	9.93	—	1.08	—	86.63	23.18	—	35.77	23.34	472.32	[39]
Shenzhen River	2014	9	12.30	0.25	0.82	—	134	178.20	—	120	92	692.20	[40]
Jialu River	2014	19	6.31	0.10	2.93	—	60.80	39.22	—	42.44	29.35	107.58	[41]
Wusuli River	2014	40	—	—	—	9.30	50.75	17.43	194.75	19.28	57.75	50.75	[42]
Yangtze River	2014	61	25.40	0.16	2.46	18.53	89.54	82	—	37.40	60	174	[43]
Wuyuer River Basin	2014	187	15.25	0.12	0.13	—	54.49	19.58	—	61.40	24.87	80.11	[44]
Lancang River	2014	22	47.33	—	—	14.40	128.23	37.38	562.91	49.38	98.27	99.67	[45]
Sanmenxia Section of Yellow River Wetland National Nature Reserve	2015	7	—	—	0.11	—	53.60	39.30	1633.50	—	41.10	72.40	[5]
Dayan River	2015	11	—	—	0.77	—	—	103.40	—	—	76.72	188.26	[46]
Yellow River	2015	—	—	—	0.23	—	77	34	912	—	27	97	[47]
Inflow Rivers of Taihu Basin	2015	71	—	0.16	0.74	15.04	165.57	115.78	503.10	63.05	69.39	344.03	[48]
Hainan Island rivers	2015	36	8.79	0.05	0.33	—	56.48	33.35	—	—	43.44	102.10	[49]
Rivers in Yangyuan	2015	—	—	—	5.71	—	138.97	87.20	—	75.07	64.47	322.40	[50]
Dan River Drainage	2015	95	7.97	—	0.80	0.81	4.69	2.13	39.76	1.91	3.40	12.76	[51]
Tiaozi River	2015	—	—	—	0.19	—	—	42.48	377.08	29.39	21.50	83.56	[52]
Liaohe River	2015	24	—	—	0.47	—	—	12.70	104.20	10.30	7.40	169.50	[53]
Daqing River System	2015	37	—	—	0.68	—	110.28	73.91	—	34.74	32.01	227.88	[54]
Rivers in Chongqing	2015	14	—	—	0.63	—	94.10	48.50	—	31.40	30.70	190	[55]
Yanghe River	2016	8	—	—	0.16	—	44.25	25.40	—	—	20.90	74.30	[56]
Nanfei River in Chaohu Basin	2016	21	12.20	0.90	3.80	—	143.20	145.40	—	45.70	70.80	869.30	[57]
Liangtan River	2016	10	—	—	—	—	75.77	—	533.30	—	186.16	226.60	[58]
Tributary of Zhedong River	2016	13	168.72	0.20	—	—	—	28.87	—	—	23.06	92.38	[59]
Lijiang River	2016	20	18.05	0.18	1.72	—	56.38	38.07	—	—	51.54	142.16	[14]
Upper Reach of the Yangtze River	2016	30	8.79	0.10	0.93	—	80.04	65.80	—	—	51.01	141.85	[60]
Huaihe River	2016	54	—	—	—	—	—	31.30	876.49	32.79	53.43	183.57	[61]
Gan River	2016	21	—	—	2.29	15.78	59.94	48	—	25.43	60.49	139.44	[62]
Ziya River	2016	28	—	—	1.31	—	460	91	—	39.10	49.70	459	[63]
Kuitun River	2016	18	—	—	—	5.43	69.55	50.27	551.14	22.32	26.19	92.06	[64]
Bortala River	2016	41	9.67	0.02	0.17	—	51.55	30.09	—	22.32	31.98	99.19	[65]
Duliujian River Drainage Basin	2016	42	68.40	—	0.60	—	62.10	142.50	—	33.9	30.10	111.40	[66]
Jiulong River	2016	39	—	0.17	0.96	14.92	93.64	83.03	1132.90	28.24	103.02	172.20	[67]
Qinghai Lake watershed	2016	6	10.32		0.10	—	32.20	11.59	409.40	15.83	14.43	43.40	[68]
Yalu River Estuary	2016	27	—	0.06	0.30	—	56.50	113.60	—	—	30.20	100.30	[69]
Minjiang River in Fuzhou	2016	—	10.02	—	0.90	—	66.62	42.33	—	—	79.14	195.57	[70]
Kuye River	2016	26	33.53	—	1.08	16.68	289.59	56.01	1471.91	—	51.92	172.86	[71]
Songhua River	2016	10	—	0.98	1.10	—	—	10.70	759	45.70	32.40	214	[72]
Xiaojiang River	2017	15	39.50	—	2.30	—	—	130.40	—	—	103.40	564.90	[73]
Jiaozhou Bay Catchment	2017	—	7.70	—	0.16	—	69.30	23.60	—	—	20.20	64.60	[10]
Jinjiang River Estuary	2017	14	—	0.49	1.59	13.10	99.90	102	1264	28.50	95.60	331	[74]
Qinhuai River	2017	35	10.78	0.25	0.61	—	79.92	44.71	—	34.60	33.39	149	[75]
Liaohe River protected Area	2017	19	9.88	—	1.20	—	35.06	17.82	—	17.73	10.57	50.24	[76]
Songhua River Harbin Region	2017	11	10.13	0.56	0.27	—	121.40	13.33	—	12.89	18.80	92.54	[77]
Xiangjiang River	2017	16	54.90	—	13.68	23.19	120.44	101.36	1805.17	57.14	214.91	443.32	[15]
Rivers of Yangzhou	2017	38	11.12	0.20	0.28	—	37.85	29.07	—	24.15	38.87	64.40	[78]
Huntai River	2017	184	—	—	0.29	—	30	34	551	23	1.44	71	[79]
Rivres in Beijing Central District	2017	42	6.01	0.67	0.29	—	63	45	277	—	31.10	—	[80]
Duliujiang River	2017	62	30.61	—	0.42	17.91	38.60	22.65	—	33.36	27.21	93.40	[81]
Xiaoqinghe Watershed	2017	—	4.37	—	6.20	—	257.79	73.35	—	56.89	250.49	418.44	[82]
Shaliuhe River	2017	56	10.40	—	0.12	—	49.10	19.70	618	24.90	18.70	68.10	[83]
Hengyang Segment of Xiangjiang River Basin	2018	8	135.20	1.19	21.66	—	54.59	112.10	1884	—	359.40	659.70	[84]
Zijiang River	2018	59	31.53	—	3	16.76	67.51	34.19	1322.89	34.66	35.68	141.90	[85]
Shima River	2018	40	—	—	1.05	—	141.48	186	—	79.88	—	528.98	[86]
River of Nantong	2018	134	15.8	0.13	2.53	—	112	53.90	—	31.20	448	869	[11]
Yongding River	2018	11	—	—	—	—	47.61	24.71	450.09	40.45	35.47	94.75	[87]
Taizihe River	2018	24	—	—	—	—	136.80	92.60	—	—	1434.25	1737.35	[88]
Liuyang River	2018	13	14.55	—	1.24	10.72	38.67	50.20	581.67	17.48	37.82	138.48	[89]
Baoxiang River	2018	10	—	—	—	—	24.9	34.26	—	13.52	13.99	55.25	[90]
Pearl River Delta	2018	323	18.23	0.10	0.84	—	55.19	42.89	—	—	44.61	135.87	[3]
Shaying River	2019	14	0.44	—	5.32	—	58.19	37.14	—	—	35.64	—	[91]
Modaomen Estuary	2019	19	—	—	1.16	20.05	124.13	34.64	—	35.22	51.85	161.8	[8]
Chunfeng Lake Estuary	2019	13	45.45	—	0.64	—	28.06	9.06	—	16.54	42.83	84.76	[17]
Guishui River	2019	—	6.81	—	0.14	10.48	50.45	17.95	631.74	21.78	22.42	66.76	[92]
Lishui River Watershed	2019	21	—	—	—	9.39	61.20	22.84	757.15	25.31	40.19	91.66	[93]
Quannan Section of Taojiang River	2019	12	15.95	1.70	9.09	—	38.94	43.09	—	—	48.72	156.80	[94]
Lanmuchang	2019	13	250.73	—	0.69	33.61	201.41	98.47	—	79.41	28.16	213.84	[95]
Rivers of Linyi	2019	12	—	—	—	—	14.12	12.25	—	9.99	11.75	21.73	[96]
Wei River	2019	12	5.44	0.03	—	—	59.17	—	—	—	45.96	79.08	[16]
Longjiang River	2019	6	46.72	—	13.92	—	—	—	—	—	139.23	472.83	[97]
Majiagou River	2019	12	—	—	0.76	—	107.37	28.05	—	17.82	26.98	358.54	[98]
Yunliang River	2019	6	—	—	1.83	—	68.19	19.46	—	8.16	32.75	861.63	[98]
River of Zigong	2019	15	19.46	0.18	0.64	—	61.95	48.62	—	33.76	29.92	165.03	[99]
River in the Baiyinnuoer lead–zinc Mining Area	2019	6	35.17	—	6.06	—	—	32.23	—	—	906.50	1432.88	[100]
Xiashan Stream	2020	13	12.68	1.05	2	—	112.76	261.88	—	46.52	93.62	332.83	[9]
Yarlung Tsangpo River	2020	67	23.70	3.26	0.16	10.25	82.29	46.01	628.24	36.73	37.05	75.53	[101]
Urban Rivers in Baoan District	2020	28	—	0.20	—	—	101.76	465.91	—	77.42	71.73	481.34	[102]
Qingshui River Basin	2020	32	10.23	0.20	—	—	41.38	—	—	—	13.99	—	[103]

“—” refers no data.

**Table 2 ijerph-18-06908-t002:** Statistics of metals concentrations (mg/kg) in river sediments and comparison with guidelines (mg/kg), background (mg/kg).

Items	As	Hg	Cd	Co	Cr	Cu	Mn	Ni	Pb	Zn	Reference
Min	0.44	0.02	0.06	0.81	4.69	2.13	39.76	1.91	1.44	12.76	This study
Max	250.73	8.67	40	251.58	460	520.42	1884	203.11	1434.25	1737.35	
Median	13.11	0.19	0.87	14.98	67.995	43.9	618	33.95	38.345	141.06	
Mean ± SD	27.36 ± 42.75	0.75 ± 1.56	2.87 ± 5.66	35.47 ± 66.47	88.71 ± 68.87	71.24 ± 90.55	751.27 ± 460.70	39.61 ± 30.38	84.06 ± 180.12	265.61 ± 311.35	
CV	156.28	206.77	197.56	187.38	77.64	127.10	61.46	76.70	214.28	117.22	
TEC	9.79	0.18	0.99	—	43.40	31.60	—	22.70	35.80	121	[104]
PEC	33.0	1.06	4.98	—	111	149	—	48.6	128	459	[104]
% of samples < TEC	25.45	46.16	54.76	—	13.64	31.25	—	25.75	47.48	44.90	
% of samples between TEC-PEC	52.73	38.46	28.57	—	65.91	61.46	—	54.55	43.43	38.78	
% of samples > PEC	21.82	15.38	16.67	—	20.45	7.29	—	19.70	9.09	16.32	
Background	11.20	0.065	0.097	12.70	61	22.60	583	26.90	26	74.20	[106]

**Table 3 ijerph-18-06908-t003:** Comparison of average contents of metals concentrations (mg/kg) in river sediments with other countries in the world.

Regions	As	Hg	Cd	Co	Cr	Cu	Mn	Ni	Pb	Zn	Reference
China	27.36	0.75	2.87	35.47	88.71	71.24	751.27	39.61	84.06	265.61	This study
Red River, Vietnam	—	—	0.35	—	85.71	83	806	38	66	127	[108]
Merang river, Malaysia	6.06	—	—	—	39.26	9.87	226.29	—	11.58	49.39	[109]
Khorramabad River, West Iran	5.80	—	—	—	169.60	49.40	636.30	76.80	19.20	87.60	[110]
Danube River, Serbia	13.89	0.80	1.69	—	—	50.93	—	—	64.92	270.40	[4]
Zarrin-Gol River, Iran	21.91	—	—	8.79	37.67	—	286.28	12.39	—	32.68	[111]
Danube, Europe	17.60	—	1.20	—	64	65.70	819	49.60	46.30	187	[112]
Tinto River, Spain	—	—	2.70	21	56	805	—	17	2330	901	[113]
Georges River, Australia	11	—	—	—	39	30	—	13	67	157	[114]
River Chenab, Pakistan	—	—	1.67	7.95	—	8.16	494	—	18.10	33.70	[115]
Awash River, Ethiopia	15.87	0.17	2.60	—	120.58	79.43	—	89.46	13.53	382.73	[116]

## Data Availability

Not applicable.

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
