# Peer review of "Concentrations, Distribution, and Pollution Assessment of Metals in River Sediments in China"

_ijerph, 2021, doi:10.3390/ijerph18136908_

Round 1
Reviewer 1 Report
Concentrations, distribution, and pollution assessment of heavy metals in river sediments in China
Guoqi Lian and Xinqing Li
The authors present in this study the concentrations, spatial distribution, and pollution assessment of heavy metals (As, Hg, Cd, Co, Cr, Cu, Mn, Ni, Pb, and Zn) in river sediments in China (the authors include As as heavy metal!).
Major comment:
The article could be of interest to the reader of the journal but has serious flaws and by no means can be published as it is.
The main reason is that the study design is not sound:
Abstract: Please give clear information on the objectives.
The retrieved results from the 102 investigations are not presented properly, and the interpretation is based only on average values, which is highly questionable as it would indicate that all heavy metals (As, Hg, Cd, Co, Cr, Cu, Mn, Ni, Pb, and Zn) in each river follow a normal distribution (or in case of median values a log-normal distribution).
In the article, a database should be presented (as attached file) including all data and analytical methods.
Not only a table with the complete results of the sediment analyses is not given, but there is also confusion in Table 1, where the statistics of trace elements in surface water and groundwater are presented.
There is no information given on how the authors did assure the quality of sediment analyses? Are they really comparable and which analytical methods and standards had been used?
The authors claim that “the results can provide scientific support for environmental management to develop corresponding control measures, and it will generate beneficial impacts on aquatic ecosystems and human health” but this would need a more precise investigation of the data.
Specific comments
Line 137 – 139
The authors state that “an inverse distance weight interpolation method (IDW) was performed to map the spatial distribution of heavy metals in rivers sediments selected in China based on ArcGIS (Version 10.5)”
The interpolation of the results, as done in the maps of Figure 3, does not make sense as the results are related only to their watersheds and are mostly independent with respect to neighboring areas.
Line 88 - 92
Comment:
This screening procedure is very basic and must be followed by an elaboration of a revised database: it is not enough to interpret only the mean contents of heavy metals, research area, river name, published year, and sampling number of the selected articles in further statistical analysis.
Line 92 - 94
Comment
The authors do not give any information on any quality control, analytical methods, or other information retrieved for the articles.
It is highly questionable that the data correspond always to normal distributions
Figure 2. River sites reviewed in this study.
The quality of the figures is very poor and data should be related to their watershed and not to China as a whole.
No information is given on the analytical methods applied in these articles. Is the concentration
How did you define the geogene background?
How did you treat values under the detection limit?
How many samples are represented at each watershed?
Minor observations:
Line 40
…, PH, …
Change into: , pH ,
Line 338
…, Mn, Ni, Pb and Zn ….
change into:
…, Mn, Ni, Pb, and Zn …
Line 342:
…, Ni and Pb showed …
Change into:
…, Ni and Pb, showed …
Line 343:
… Hg, Cd, Cu and Zn.
Change into:
… Hg, Cd, Cu, and Zn.
Lines 341 - 343
The Igeo values suggested that most sites of As, Co, Cr, Mn, Ni and Pb showed uncontaminated, while it existed most proportions of contamination in Hg, Cd, Cu and Zn.
What do you mean by “showed uncontaminated, while it existed most proportions of contamination in Hg, Cd, Cu and Zn” ?
Comment: Please rewrite
Lines 343 - 344
Among these metals, Cd and Hg presented higher proportions of contaminated sites, …
Comment: Please rewrite
Line 344 -346
..and contributed most to the potential ecological risk in the river sediments, could be regarded as dominant pollutants among the heavy metals in this study.
Change into:
..and contributed most to the potential ecological risk in the river sediments, which could be regarded as dominant pollutants among the heavy metals in this study.
Line 351 - 352
The study provided certain theoretical basis to strengthen the management of pollutant discharge in the river basins and take reasonable protection measures for the …
Comment: Please rewrite
Line 353 - 354
… such as Hg, Cd and As.
Change into:
… such as Hg, Cd, and As.
Line 354 - 356
Furthermore, future studies should carry out detailed investigation and research on pollution sources in different river basins and formulate pollution control measures, according to different basins.
Comment: Please rewrite
Author Response
The detail was presented in the document, please check the attachment.

Reviewer 2 Report
The present manuscript describes the contamination status of ten heavy metals in riverine sediments collected from several rivers in China. With the generated data and data obtained from literature survey, the authors conducted assessment of heavy metal pollution using ecoltoxicological thresholds and geoaccumulation index. The major findings are to summarize the digital values of heavy metals in riverine sediments from China and to show spatial distribution of metals. The assessment results are providing that certain metals are exceeding threshold values.
I think the present manuscript looks not acceptable to IJERPH journal because the study is providing limited original data on heavy metals in sediments. Although the authors summarized water contamination by heavy metals, it was not discussed for implications. I think multiple environmental matrix could provide more reliable insights for metal pollution. The authors tried to identify metal pollution in riverine sediment but it looks not clear without any analytical methods as well as quality control data. I am not sure the present manuscript belongs to review paper or original research paper. To clarify this, the authors should make a decision how to organize the present manuscript with a focus on types of paper. In terms of review paper, it looks not acceptable novelty for pollution assessment of heavy metals, but it could be resubmitted for publication after revision of specific comments below as;
Specific comments
Title
- I think authors can change the title from ‘river sediments’ to ‘riverine (or freshwater) environments’ because the authors dealt with water and sediment concentrations from riverine environments for a comprehensive review. Thus, to express more wide ranges of investigations conducted in the manuscript, title could be changeable.
Abstract
- Please make consistent significant digit of value for heavy metals.
- The more explanations are needed for why the eastern part of China is highly contaminated by heavy metals.
Introduction
- Line 38: I am not sure that the word ‘non-degradable and biomagnification of metal’ is right. Maybe methylated mercury could be possible for biomagnification. Please check and rephrase the sentence like less degradable and bioaccumulation.
- Chang the study objectives because the present study did not measure heavy metals and only investigated heavy metal contamination with literature survey. Please clarify this. If the authors measured the heavy metals in riverine sediments, please describe how to measure targeted contaminants in ‘M&M’.
Materials and methods
- SQGs of sedimentary metals should be provided with what kinds of ecotoxicological endpoints and tested organisms for meaningful interpretations.
- Very confusing on search method. If the authors measured heavy metals in sediment collected from several rivers, why did the authors use ‘searching method’? only for comparison?
Table 1
- Add sampling year and numbers of sediment samples in Table 1 and then sort heavy metal data following the sampling years for investigating time trends of heavy metals
- Please make consistent significant figures of all data
Tables 2 and 3
- Please make consistent significant figures of all data
Figure 3
- I am not sure how all data were combined for this figure? How did authors consider different sampling years for contour figure of heavy metal contamination.
Figures 4 and 5
- I am wondering what is the original data generated from the manuscript? If the authors merge all data set, it could make a confusion.
Round 2
Reviewer 1 Report
I still have great problems with the manuscript, as the main problems have not been solved. I don’t think the article may be published in the present form. It must be based on a reliable database that should be attached to the article
Point 1: Abstract: Please give clear information on the objectives.
Response 1: It has been corrected.
OK
Point 2:The retrieved results from the 102 investigations are not presented properly, and the interpretation is based only on average values, which is highly questionable as it would indicate that all heavy metals (As, Hg, Cd, Co, Cr, Cu, Mn, Ni, Pb, and Zn) in each river follow a normal distribution (or in case of median values a log-normal distribution).
Response 2: The mean values of the metals in 102 investigationswere obtained from the literature, and the standard deviation (SD) and variable coefficient (CV) couldn’t get from most of the investigations. We have added the SD and CV values of the total metal in Table 2 and increased the results in the manuscript.
This is a problem as it means that you do not have access to the original data.
Point 3:Not only a table with the complete results of the sediment analyses is not given, but there is also confusion in Table 1, where the statistics of trace elements in surface water and groundwater are presented.
Response 3:The values of some metals among the 10metals were not all presented in the literature, the statistics of metals were based on the surface water.
Is this article about water quality or about soil quality (as indicated in the title?) You cannot substitute one with the other.
Point 4:There is no information given on how the authors did assure the quality of sediment analyses? Are they really comparable and which analytical methods and standards had been used?
Response4:The sample analysis process was added in the manuscript.
OK
Point 5: Line 137 – 139,The authors state that “an inverse distance weight interpolation method (IDW) was performed to map the spatial distribution of heavy metals in rivers sediments selected in China based on ArcGIS (Version 10.5)”
The interpolation of the results, as done in the maps of Figure 3, does not make sense as the results are related only to their watersheds and are mostly independent with respect to neighboring areas.
Response5:TheIDW or kriging interpolation can reveal the spatial distribution of the metals in the watershed, according to the different colours of the values, and the IDW can better show the distribution compared with the kriging interpolation in this study, so we selected the IDW.
I would prefer to see symbols indicating different concentrations rather than a spatial distribution as information is limited to sediments in riverbeds and therefore cannot be interpolated through the whole of China!
Point 6:Line 88 – 92,This screening procedure is very basic and must be followed by an elaboration of a revised database: it is not enough to interpret only the mean contents of heavy metals, research area, river name, published year, and sampling number of the selected articles in further statistical analysis.
Response6:The year and number of the sample sites were added in Table, and the SD and CV were added in Table 2, the relevant content has been added.
I would expect that your analysis was based on a complete data table, which is not the case.
Point 7:Line 92 – 94,The authors do not give any information on any quality control, analytical methods, or other information retrieved for the articles.
It is highly questionable that the data correspond always to normal distributions.
Response7:The sample analysis process was added in the manuscript.
OK
Point 8:Figure 2. River sites reviewed in this study.
The quality of the figures is very poor and data should be related to their watershed and not to China as a whole.
Response8:The data were not related to their watershed for the uneven distribution of sample points, in some watershed they were too few.
This problem is still not solved
Point 9:
No information is given on the analytical methods applied in these articles.
Response 9: It has been added.
OK
Point 10:
How did you define the geogene background?
How did you treat values under the detection limit?
How many samples are represented at each watershed?
Response10:
The backgroundwas based on the Background values of elements in soils of China (CEMS,1990).
OK
The values under the detection limit were not taken into account.
How can you then calculate the mean value of only one part of the analyses? This will lead to an overestimation!
The sample sites in different river basins were added in the “Searching Method” .
Point 11:
Line 40
…, PH, …
Change into: , pH ,
Response 11:It has been corrected.
OK
Point 12:
Line 338
…, Mn, Ni, Pb and Zn ….
change into:
…, Mn, Ni, Pb, and Zn …
Response 12:It has been corrected.
OK
Point 13:
Line 342:
…, Ni and Pb showed …
Change into:
…, Ni and Pb, showed …
Response 13:It has been corrected.
OK
Point 14:
Line 343:
… Hg, Cd, Cu and Zn.
Change into:
… Hg, Cd, Cu, and Zn.
Response 14:It has been corrected.
OK
Point 15:
Lines 341 - 343
The Igeo values suggested that most sites of As, Co, Cr, Mn, Ni and Pb showed uncontaminated, while it existed most proportions of contamination in Hg, Cd, Cu and Zn.
What do you mean by “showed uncontaminated, while it existed most proportions of contamination in Hg, Cd, Cu and Zn”?
Response 15:The results indicated that metals of As, Co, Cr, Mn, Ni and Pb showed uncontaminated,while other metals such as Hg, Cd, Cu and Znexisted certain proportions of contamination.
OK
Point 16:
Lines 343 - 344
Among these metals, Cd and Hg presented higher proportions of contaminated sites, …
Comment: Please rewrite
Response 16:It has been corrected.
OK
Point 17:
Line 344 -346
..and contributed most to the potential ecological risk in the river sediments, could be regarded as dominant pollutants among the heavy metals in this study.
Change into:
..and contributed most to the potential ecological risk in the river sediments, which could be regarded as dominant pollutants among the heavy metals in this study.
Response 17:It has been corrected.
OK
Point 18:
Line 351 - 352
The study provided certain theoretical basis to strengthen the management of pollutant discharge in the river basins and take reasonable protection measures for the …
Comment: Please rewrite
Response 18:It has been corrected.
OK
Point 19:
Line 353 - 354
… such as Hg, Cd and As.
Change into:
… such as Hg, Cd, and As.
Response 19:It has been corrected.
OK
Point 20:
Line 354 - 356
Furthermore, future studies should carry out detailed investigation and research on pollution sources in different river basins and formulate pollution control measures, according to different basins.
Comment: Please rewrite
Response20:It has been corrected.
OK
Reviewer 2 Report
I think the present manuscript was well revised with comments and suggestions from the reviewer. I would like to recommend the revised manuscript acceptable to IJERPH.
Author Response
Thank you!